# Garcinol—A Natural Histone Acetyltransferase Inhibitor and New Anti-Cancer Epigenetic Drug

**DOI:** 10.3390/ijms22062828

**Published:** 2021-03-11

**Authors:** Patrycja Kopytko, Katarzyna Piotrowska, Joanna Janisiak, Maciej Tarnowski

**Affiliations:** Department of Physiology, Pomeranian Medical University in Szczecin, al. Powstancow Wlkp. 72, 70-111 Szczecin, Poland; patrycja.kopytko@op.pl (P.K.); joanna.janisiak@gmail.com (J.J.)

**Keywords:** garcinol, HAT inhibitors, epigenetic drugs, miRNA, cancer

## Abstract

Garcinol extracted from *Garcinia indica* fruit peel and leaves is a polyisoprenylated benzophenone. In traditional medicine it was used for its antioxidant and anti-inflammatory properties. Several studies have shown anti-cancer properties of garcinol in cancer cell lines and experimental animal models. Garcinol action in cancer cells is based on its antioxidant and anti-inflammatory properties, but also on its potency to inhibit histone acetyltransferases (HATs). Recent studies indicate that garcinol may also deregulate expression of miRNAs involved in tumour development and progression. This paper focuses on the latest research concerning garcinol as a HAT inhibitor and miRNA deregulator in the development and progression of various cancers. Garcinol may be considered as a candidate for next generation epigenetic drugs, but further studies are needed to establish the precise toxicity, dosages, routes of administration, and safety for patients.

## 1. Introduction

The term ‘epigenetics’ was originally defined by Conrad Waddington to describe heritable changes in the cellular phenotype that are independent of changes in the DNA sequence [1]. The current definition of epigenetics is the study of heritable changes in gene expression that occur independently of changes in the original DNA sequence. Epigenetic modifications play a fundamental role in many aspects of cell development, from embryogenesis to the determination of cell fate and lineage. Epigenetics also plays an important role in the modulation of genetic diversity, such as phenotypic variation among genetically identical individuals [2,3]. Many types of epigenetic modification have been recognised, including methylation, acetylation, phosphorylation, ubiquitination, sumoylation, ribosylation, and deimination. Epigenome imbalances can over-activate or inhibit multiple signalling pathways, leading to the development of many diseases [4,5].

Acetylation is one of the most important and most common post-translational modifications of proteins, which modulates transcription, chromatin structure, and DNA strand repair [6]. This highly dynamic process induces conformational changes to the nucleosome with consequent activation of transcription. The enzymes that play a key role in the acetylation process are histone acetyltransferases (HATs), otherwise known as K-acetyltransferases (KATs) [7,8].

## 2. Histone Acetyltransferases

Based on cellular localisation, HATs can be classified into two separate groups with distinct functions. Type A acetyltransferases are located in the cell nucleus and catalyse transcription processes [9,10]. They contain a bromodomain that recognises and binds to acetylated lysine residues on histone substrates. This group includes P300/CBP-associated factor (PCAF), general control non-repressed 5 protein (GCN5), CREB-binding protein (CBP), histone acetyltransferase p300 (P300), tat-interactive protein-60 (TIP60), human acetyltransferase binding to ORC (HBO1), and steroid receptor coactivator-1 (SRC1) [11,12,13]. The activity of B-type cytoplasmic acetyltransferases is the acetylation of newly synthesised histones; additionally, they can catalyse the acetylation of non-histone proteins. In contrast to type A HATs, type B HATs do not have a bromodomain; an example of an enzyme belonging to this subgroup is histone acetyltransferase 1(HAT1) [9,11,14].

HAT enzymes are highly diverse; therefore, they can be grouped based on the presence of catalytic domains into four major families: GNAT, MYST, P300/CBP, and steroid receptor coactivators (SRC) (Figure 1) [15].

The first and most numerous group of HATs is the MYST family. It includes the enzymes HBO1, MOF, TIP60, MOZ, and MORF. HATs belonging to the MYST family play a key role in the post-translational modification of histones by influencing chromatin structure in the eukaryotic nucleus. They are characterised by the highly conserved MYST domain composed of an acetyl-CoA binding motif and a zinc finger. Some members of this family also share additional structural features, such as chromodomains or zinc fingers linked to the plant homeodomain (PHD) [11,16]. It has been shown that they act as catalysts for the acetylation of lysine residues in histones H2A, H3, and H4 and use a ping-pong catalytic mechanism [17]. 

The GNAT subgroup of acetyltransferases includes GCN5, PCAF, HAT1, and ELP3. They have a bromodomain in their structure and catalyse the acetylation of a lysine residue on histones H2B, H3, and H4. They use a ternary complex catalytic mechanism, and the amino and carboxyl-terminated segments are used to bind the histone substrate [17].

The P300/CBP family of acetyltransferases includes P300 and CBP and use the Theorell–Chance (“hit-and-run”) ternary catalytic mechanism. They contain a substrate binding loop that participates in the binding of AcCoA and lysine, and an autoacetylation loop that requires autoacetylation of lysine to achieve maximum catalytic activity [18]. P300/CBP transcription coactivator proteins play a key role in coordinating and integrating many events that are dependent on the transcriptional apparatus, enabling the appropriate level of gene activity to occur in response to various physiological signals that affect proliferation, differentiation, and apoptosis [19].

The least common, and least well-known, family of HATs are steroid receptor coactivators, represented by SRC1, ACTR, and TIF2. They have the intrinsic activity of HATs and interact strongly with P300/CBP and PCAF. Steroid receptors and coactivator proteins are believed to stimulate gene expression, facilitating the accumulation of essential transcription factors into a stable pre-initiation complex [20].

HATs mediate a wide variety of biological processes, including cell cycle progression, X-chromosome dosage compensation, hormonal signalling, and DNA damage repair. Abnormal functioning in HATs has been correlated with some diseases, including cancer, pneumonia, viral infections, diabetes, fungal infections, and drug addiction [21,22].

P300/CBP and PCAF Acetyltransferases as Key Targets for Epigenetic Treatment

The abnormal expression of HATs leads to a variety of defects in the cell, for example, excessive proliferation or metastasis. Elevated levels of P300/CBP and PCAF have been reported in many malignancies. Studies with selective P300 and CBP inhibitors or silenced expression revealed different routes of P300/CBP action in cancer development and progression (Table 1).

In melanoma cell lines, the silencing of P300 expression downregulated hundreds of genes. Many of them were connected to thecell cycle, i.e., cyclin A2. In contrast, p21 cyclindependent kinase inhibitor (CDKN1A) was upregulated [23]. Other transcriptional factors or oncogenes become overexpressed due to histone acetylation by P300/CBP [23,24,25,26]. 

**Table 1 ijms-22-02828-t001:** Involvement of P300/CBP HAT in cellular pathways.

Pathway/ Element	Pathway Role	Cell Line	Inhibitor/Silencing of p300 or CBP	Reference
↓Cyclin A2 ↑CDKN1A	Cell cycle	Human melanoma	Silencing	[27]
↑NKG2D-L	Innate immune response NK cells	HUVEC,911, WI-38, SKOV3, HepG2, MCF7, MDA-231	Inhibition (C646)	[28]
↓PD-L1	Innate immune response CD8+ T cells	Prostate cancer cell lines: DU145, Pc3, 22Rv1, LNCaP	Inhibition (A485)	[29]
↓PIK3R1/P50	Cell proliferation, differentiation and survival	Primary breast cancer patient samples, cell lines MCf7, MDA231, ZR75-1, T47D, nude balb/c mice	Not applicable	[30]
E2F1	Double strand break repair	MEFs from FVB mice (E2f1Knock in)	Not applicable	[31]

HAT—histone acetyltransferase, NK—natural killer cells.

The inhibition of P300/CBP in cell lines showed that its involvement in cancer progression is beyond the cell cycle and apoptosis. Early stages of cancer development require escape from the immune system’s early response (innate immunity). Natural Killer group 2 member D ligand (NKG2D-L) is a protein not found in normal cells but rather in cells that are extensively proliferating, such as those during wound healing, in inflammatory disorders, and in cancer cells. NKG2D-L is recognised by Natural Killer (NK) cells, which causes lysis of ligand positive cells [32]. P300/CBP inhibits the upregulation of NKG2D-L in vitro and in vivo. The inhibition of P300/CBP by selective inhibitors restores high expression of NKG2D-L and allows NK cell-mediated lysis of cancer cells [32]. The response of the immune system on carcinogenesis is also pronounced by CD8+ T cells. Programmed-death-ligand (PD-L1), which blocks T cell function, is upregulated in some cancers. PD-L1 is found in exosomes and increases its expression with cancer progression in human cancer samples [33]. Increased PD-L1 is correlated with increased CBP and P300. The inhibition of P300/CBP by the synthetic A-485 inhibitor has decreased PD-L1 in cells and exosomes. In mice, A-485 decreased cancer cell proliferation and increased T cell infiltration without significant weight loss, which indicates low toxicity of A-485 [33].

Chi et al. showed that the CapG protein mainly influences actin filaments, thus regulating cell differentiation, phagocytosis, and motility [26]. When acting with the P300/CBP complex, it increases PIK3R1/P50 expression, which is required for PI3K action. Most human cancers exhibit PI3K pathway activation, and a variety of aberrations are noted in this pathway in human cancers (reviewed in [27]). The PI3K route is also linked to cancer resistance to chemotherapeutic agents [26]. 

Another type of resistance is the increased ability to repair DNA damage. Double strand breaks (DSB) in DNA increase radiotherapy efficiency, and resistance to this type of therapy is related to effective DNA repair in cancer cells. The E2F1 transcription factor is responsible for the regulation of proliferation and apoptosis and plays a role in DNA repair; it is deregulated in many types of cancers. E2F1 may be phosphorylated and acetylated in cancer cells. Both modifications are needed in DNA repair for the localisation (phosphorylation) and gathering (acetylation) at thesite of the DNA break. In the acetylated form, E2F1 recruits P300/CBP to the site of DNA damage, causing chromatin remodelling and facilitating DNA repair [34]. Drug-resistant PTEN^−/−^ cancer cells treated with anacardic acid (AA) showed decreased viability and increased apoptosis [35]. In this experiment, the authors also observed Heat Shock Protein 70 (HSP70) and AKT1 reduction. Proper functioning of the cell depends on proper protein folding. HSP70 is responsible for folding and refolding misfolded proteins and for the degradation of abnormal proteins. Downregulation of P300/CBP and the reduction of HSP70 causes AKT1 reduction on a transcriptional and post-translational level in PTEN^−/−^ prostate cancer cells [35].

Increased PCAF and H3S28ph expression was noted in osteosarcoma cell lines compared to osteoblastoma cell lines. PCAF overexpression causes high expression of H3S28ph, evidencing a regulatory relationship between PCAF and H3S28ph. A study showed that PCAF interacts directly with H3, and the silencing of PCAF expression results in a reduction in autophagy and viability in osteosarcoma cells. The inhibition of autophagy by silencing PCAF is associated with lowering the expression of autophagy-related proteins (ATG): ATG5, ATG13, and ATG14 [30].

One of the human mucins, MUC1, belongs to the factors promoting cancer progression. Fernandez et al. investigated the effect of MUC1 overexpression in colon cancer cells and found that it stimulates EMT and increases the migration, invasion, and phosphorylation of AKT. Use of the aspirin metabolite salicylate, which is essential for the activity of AA, decreased the level of MUC1, inhibited EMT, and decreased AKT phosphorylation. Therefore, it has been hypothesised that these effects are related to the inhibition of HATs (Table 2). In vitro tests confirmed that salicylate directly inhibited the activity of PCAF, TIP60, and MOF in PC-3 prostate cancer cells, probably by reversing the epithelial–mesenchymal transition [28].

It has been proven that the strategy of HAT blocking using bi-substrate inhibitors is a very effective way of obtaining high affinity and selectivity of action by bioactive compounds. Their structure resembles substrates of HATs with an acetyl-CoA and a peptide resembling a lysine substrate [15,36].

Synthetic inhibitors may play a key role in the treatment of oncological diseases in the future. Currently, the search for such compounds is laborious. The starting points for the development of more potent and highly specific inhibitors of HATs are high-throughput screening and in silico library screening, as well as ligand-based design. Therefore, a thorough screening of thiazole, isothiazole, and nitrogen cyclic compounds, among others,was carried out and then modified to obtain improved HAT inhibitors [15,31,37].

Natural products are usually important starting points for the development of new drugs. Polyphenols that are common in plants have been studied as inhibitors of HAT activity. Because polyphenols have bioactive properties, attempts have been made not only to improve their activity but also their selectivity and stability [15,21]. This review focuses on HAT inhibitors and garcinol as natural substances with epigenetic and anti-cancer properties.

## 3. Garcinol

Although the properties of dried plum extract of *Garcinia indicia* are known in traditional Chinese medicine, its chemical structure was described by Sahuet in 1989 (Figure 2) [38].

**Figure 2 ijms-22-02828-f002:**
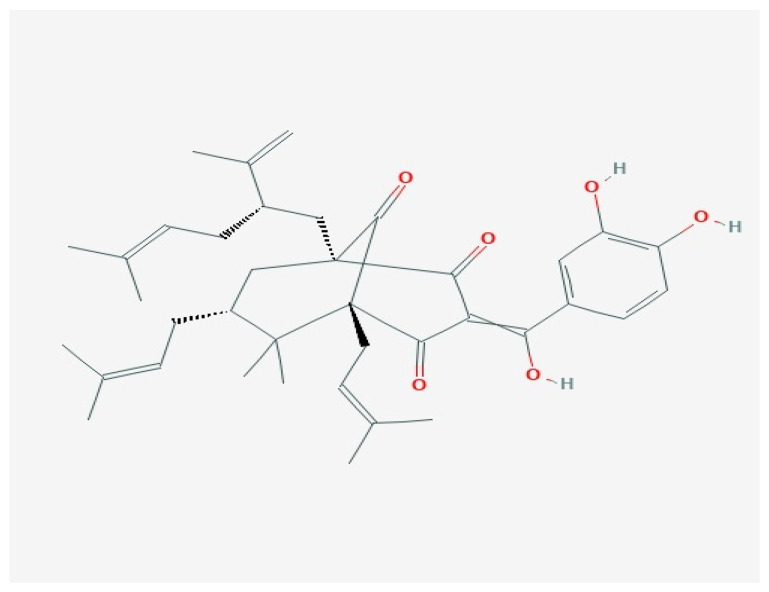
Garcinol structure. Accessed from pubchem.ncbi.nlm.nih.gov [39].

Other potential sources of garcinol are *G. cambogia, G. morella, G. yunnanesis, G. xantochymus, G. travancorica,* and *G. Buchananii* [38]. Garcinol can be obtained by methanol extraction followed by multi-step evaporation and absorption reactions (column chromatography absorption; reviewed in [40]). Dried fruit of *G. Indica* contains around 2.5% garcinol when measured with High-Performance Thin Layer Chromatography (HPTLC). Studies of garcinol’s chemical structure revealed that the effects of this substance are the result of specific chemical groups in its molecular structure. The C-3 ketonic group and phenolic ring with hydroxyl attachments are the oxidation sites. 1,2-Carbon double bond α,β-unsaturated ketones have been found to be crucial for apoptosis induction. Double bonds in the isoprenyl ring are responsible for the antioxidative properties of garcinol, while 13,14-dihydroxy groups and C8 side chains are key functional groups for anti-cancer effects as shown in in vitro experiments [41,42]. Recently, garcinol was shown to exert anticancer effects on cancer cell lines, in vitro cultures, and in vivo experiments in animal models (Table 3) [38,43].

### 3.1. Anti-Cancer Properties of Garcinol 

Garcinol exerts its anticancer effects by affecting multiple intracellular pathways. An experiment on HeLa cells revealed global change in gene expression after garcinol treatment. Balasubramanyam and coworkers estimated that about 6 to 8% of known genes were up or downregulated after garcinol stimulation [43]. Many of them were related to apoptosis, cell cycle, transcriptional regulation, and oncogenesis [43].

Further experiments showed the potency of garcinol as an inhibitor of proliferation of cancer cells. This activity was confirmed in HeLa cells, ovarian cancer cells, HT-29 (human colon cancer cells), gastric cancer cells, oral squamous cell carcinoma, glioblastoma, melanoma, pancreatic cancer, and endometrial cancer. Garcinol affected cell cycle progression and transition between G1 and S phases, resulting in increased amounts of cells arrested in G0/G1 phase and a decreased number of cells in S phase. Garcinol decreased cyclin D1 and cyclin D3 expression, decreased STAT expression, and caused inhibition of the PI3K/AKT pathway, which is crucial for proliferation, invasion, and metastasis [37,44,45,46]. In H1299 lung cancer and endometrial cancer cell lines, garcinol application resulted in cell cycle arrest and a marked decrease in cyclin-dependent kinase 2 (CDK2) and cyclin-dependent kinase 4 (CDK4) expression [44,46].

Increases in Bax and PARP (proapoptotic) expression and a decrease in Bcl-2 (anti-apoptotic) expression together with activation of caspases 3 and 9 was noted after garcinol treatment [38,47,48]. Combined treatment with garcinol and TRAIL (TNF-related apoptosis-inducing ligand)caused cancer cell apoptosis [49]. Renal cancer cells, lung cancer cells, and hepatoma cells showed increased expression of death receptor (DR5) and downregulation of c-Flip protein expression after co-treatment, but viability of normal cells remained unchanged [49]. 

Other anti-cancer properties of garcinol are connected with inhibition of metastasis, migration, and colony/sphere formation in in vitro and in vivo studies. Cells characterised by high invasive potential are the result of epithelial–mesothelial transition (EMT) during which the cells lose polarity and adhesiveness, characteristic to epithelial cells, and become mesenchymal fibroblast-like cells: nonpolar and with high motility [50].It was confirmed that garcinol treatment decreases metalloproteinases (MMP) 2 and 9 expression [37,51,52,53]. In experiments with a gastric cancer cell line and hepatocellular carcinoma, STAT3 and AKT phosphorylation was also decreased [37,54]. Recently, the inhibition of NF-κB/Twist-related protein (TWIST1) was reported as a result of synergic taxol and garcinol treatment [55]. In a study on aggressive triple negative MDA-MB-231 and BT549 breast cancer cells, garcinol induced mesothelial-to-epithelial transition (MET) by upregulation of miRNAs and other factors responsible for this process [56].

Garcinol exerts beneficial effects not only in differentiated cancer cells but also oncancer stem cells. The presence of cancer stem cells in tumour increases tumourigenic properties and chemotherapeutic resistance, which is associated with poor prognosis [57,58]. Recent studies have shown decreased self-renewing abilities and suppressed tumour sphere formation in non-small lung cancer cells (A549 cell line), by decreasing expression of ALDH1A1, and in pancreatic cancer cells (PANC-1-SP) by inhibition of Notch1 expression [57,58].

### 3.2. Garcinol as an Effective Inhibitor of HATs and Putative Epigenetic Drug

Balasubramanyam et al. proved that garcinol is a potent inhibitor of P300 and PCAF HATs. In order to test the nature and mechanism of inhibition, researchers measured the rate of acetylation of core histones in HeLa cells, in the presence or absence of garcinol and fixed or various concentrations of [^3^H] acetylCoA. Garcinol was shown to act as a competitive inhibitor that competes with histones to bind to the active site of the enzyme. It inhibits histone acetylation in vivo but has no effect on histone deacetylation. Interestingly, garcinol inhibits P300-driven histone H4 acetylation more effectively and faster than histone H3. In addition, the analysis of expression microarrays showed that it globally suppresses transcription [43].

Garcinol has also been noted to cause histone H3K18 hypoacetylation in breast cancer cells [59]. The inhibition of CBP/P300-mediated acetylation of H3K18 may be a contributing factor to the failure of MCF7 cells to transition to phase S of the cell cycle. In contrast, it was not found to significantly affect H3K9 acetylation. This reaction is catalysed by GCN5, which may suggest its insensitivity to garcinol. In addition, it has been shown that garcinol exerts other biological effects on cancer cells, including activation of the signalling pathway associated with DNA damage and the induction of chromatin regulators, such as TIP60 and SUVOH2 [59].

Garcinol induces P300 degradation by affecting the lysosomal pathway [60]. P300 acetyltransferase is involved in the regulation of FOXP3, and the use of garcinol leads to hypoacetylation and degradation of FOXP3. Garcinol treatment is associated with decreased T_reg_ suppressive activity, inhibition of tumour growth, and improved anti-tumour activity of targeted anti-p185^her2/neu^antibody therapy. The above research may form the basis for the development of new cancer treatment strategies involving T_reg_ cells [60].

A study using oesophageal cancer cells, KYSE150 and KYSE450, was aimed at checking whether garcinol can inhibit the metastatic process and determining the mechanism of its action [61]. It was noted that garcinol inhibits migration and invasion in wound healing, Transwell, and Matrigel assays in a dose-dependent manner. Moreover, it has been proven that garcinol lowers the level of P300 and CBP proteins in a dose-dependent manner. Additionally, increased activity of the epithelial mesenchymal transition-related protein E-cadherin and down regulated vimentin and SNAI1 (snail) was observed. In in vivo studies of a pulmonary metastasis mouse model, it was shown that both garcinol and 5-FU reduced the number of lung tumours compared to vehicle-only control animals. In addition, after the administration of garcinol and 5-FU, Ki-67 expression was significantly reduced, together with reduced levels of P300 and p-Smad2/3 in lung tissues. This study showed that garcinol affects P300 and TGF-β1 pathways in oesophageal cancer cells [61]. ADA3 protein is an essential adaptive component of several lysine acetyltransferase (KAT) complexes involved in chromatin modifications, playing a key role in cell cycle progression and the maintenance of genome stability. A study by Srivastava et al. showed that garcinol causes proteasome-dependent destabilisation of the ADA3 protein in HER2+ breast cancer cells. Reducing the level of ADA3 and P300 phosphorylation affects the inhibition of the cell cycle and apoptosis. Garcinol affects the RTK-AKT-P300-ADA3 signalling pathway, which is accompanied by the accumulation of p27, a decrease in the mitotic marker pH3 (S10), a reduction in the level of the S-PCNA phase marker, and induction of PARP [62].

The anticancer effects of garcinol are the result of the influence of this substance on many different pathways. The experiment on HeLa cells revealed global changes in gene expression after garcinol treatment. Balasubramanayam et al. estimated that 6 to 8% of known genes were up- or downregulated after garcinol stimulation in HeLa cells [43]. Many of them were related to apoptosis, the cell cycle, transcription factors, and oncogenes [43].

### 3.3. Garcinol Affects miRNA Expression

The epigenetic action of garcinol is not restricted to HAT inhibition. Recent studies indicated the influence of garcinol on miRNAs (Table 4). miRNAs are involved in the regulation of cell proliferation, migration, and apoptosis by controlling the expression of suppressor genes and oncogenes. In traditional mechanisms of action, miRNA binds to the target genes and regulates its expression. Recently, new ‘non-canonical’ models of action were proposed for regulation through the recruitment of other proteins, influence on ribosome biogenesis, or by skips in the cell cycle [63]. About 50% of miRNAs are located at sites in the genome frequently amplified or deleted in cancers [64].

**Table 4 ijms-22-02828-t004:** Garcinol effects on miRNA in cancer cell lines.

miRNA	Cell Type	Effect of Garcinol	Reference
miR-200b, miR-200c,let-7	MDA-MB-231 BT-549 breast cancer lines	Upregulated- reversed EMT to MET	[56]
	PANC-1-SP pancreatic cancer cells	Upregulated- caused ↓Notch1 and ↓Oct4	[58]
	A549 chemo-resistant lung cancer line	Upregulated- reversed EMT	[65]
miR-218, miR-101, miR-205	A549 chemo-resistant lung cancer line	Upregulated-reversed EMT at a lesser degree than miR-200 and let-7	[65]
miR-181	glioblastoma	Upregulated-caused ↓STAT, ↓migration	[66]

Two groups of differentially regulated miRNAs were revealed in PaCa cells treated with garcinol alone and in combination with gemcitabine [51]. Tumour suppressors were upregulated and oncogenes were downregulated by garcinol. Synergistic effects were observed in treatments with gemcitabine. Moreover, miR-21, which is related to gemcitabine resistance, was downregulated [51].

Deregulation of miR-200s and let-7 was shown in triple negative breast cancer cell lines, where treatment with garcinol caused MET and decreased invasive potential in in vitro and in vivo experiments [56]. Changes in miR-200c influenced self-renewal abilities in stem cells (PANC-1-SP line) by influencing Notch1 and Oct4 expression [58]. In cancer, the inhibition of miR-181 improves migration and may be used as an independent predictor of clinical outcomes in patients with different types of cancer. Studies on glioblastomas revealed increased miR-181 expression, decreased STAT3, and diminished vimentin after garcinol treatment [66]. In chemoresistant A549 cell lines, garcinol increased apoptosis and upregulated the miRNAs responsible for the inhibition of EMT [65]. The highest effectiveness was found for miR-200b and let-7. The effect of garcinol was confirmed in both studies by specific inhibition of miRNAs, which attenuated the effects of garcinol in studied models [65,66]. Research on miRNAs with garcinol treatment is limited. Conversely, there is much data concerning the role of miRNAs in carcinogenesis and cancer progression [34]. It may be expected that research on garcinol in miRNA regulation will be continued.

Garcinol was tested as a P300/CBP and PCAF inhibitor, but there are not many studies showing the exact routes of p300/CBP action, which is inhibited by garcinol [43,59,60]. Other studies with synthetic (A-485) or natural (AA) inhibitors suggest that garcinol may act in a similar way.

## 4. Conclusions

HATs are involved in the pathogenesis of many diseases, including oncological diseases. There are many preclinical studies on the inhibition of histone acetyltransferases. Research confirms that P300/CBP and PCAF may prove to be attractive targets of epigenetic therapies. Garcinol is one compound that inhibits the activity of P300/CBP and PCAF acetyltransferases. However, the development of garcinol as a therapeutic agent requires further study. It is essential to establish toxicity, dosage, route of administration, and bioavailability under physiological conditions of the human body. Plant extracts containing garcinol have been used for centuries and are considered safe; however, it is still a challenge to perform toxicological assays with a pharmaceutically clean form of garcinol.

## Figures and Tables

**Figure 1 ijms-22-02828-f001:**
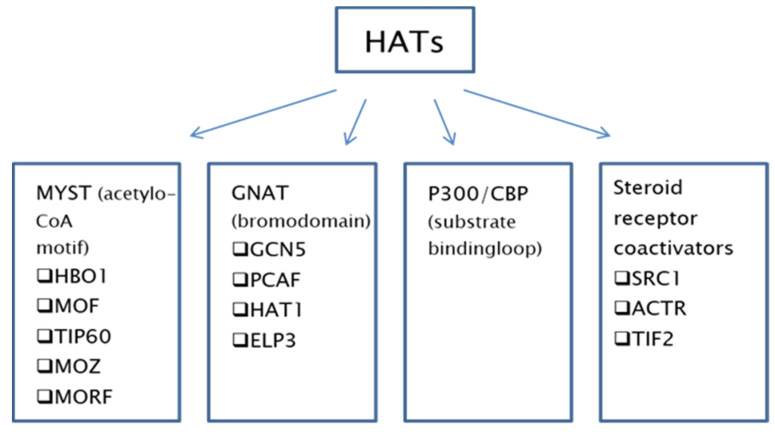
Differentiation of histone acetyltransferase subtypes based on the presence of catalytic domains.

**Table 2 ijms-22-02828-t002:** Known HAT inhibitors [22,29].

Inhibitor	Inhibited HAT
Bi-substrate inhibitors	Non-selective
Garcinol	P300
Curcumin	P300
Anacardic acid	Non-selective
TH1834	TIP60
Benzylidene barbituric acid	P300
Isothiazolones	Various
Thiazinesulfonamide	P300
C646	P300
ICG-001	CBP/β-catenin
Ischemin (bromodomain inhibitor)	Gcn5, PCAF, p300/CBP
Cyclicpeptide bromodomain inhibitors	Targets p53
N1-aryl-propane-1,3-diamine derivatives (bromodomain inhibitors)	Targets HIF-1
A-485	P300/CBP (high selectivity)

**Table 3 ijms-22-02828-t003:** Major anticancer effects of garcinol observed in in vitro studies. Adopted from Aggarwal 2020.

Effect	Type of Cancer (Cell Line)	Garcinol Concentration
Increased apoptosis	Melanoma, glioblastoma, cervical cancer, breast cancer, leukaemia, lung cancer, hepatocellular carcinoma, pancreatic cancer, colon cancer, prostate cancer	2.5–50 μM
↑caspase-3,↑caspase-9	Melanoma, leukaemia, hepatocellular carcinoma, pancreatic cancer, colon cancer	0–50 μM
Cell cycle arrest, ↓cyclins B,D1, D3, and E	Breast cancer, lung cancer, hepatocellular carcinoma	0–50 μM, 500ppm
↑Bax, ↑Bad, ↓Bcl-2, ↓Bcl-xl	Melanoma, glioblastoma, breast cancer, leukaemia, hepatocellular carcinoma, colon cancer	0–50 μM
↓NF-κBsignalling pathway	Oral squamous cell carcinoma, breast cancer, pancreatic cancer, prostate cancer	0–50 μM
↓MMP2,↓MMP9	Breast cancer, pancreatic cancer, gallbladder cancer, colon cancer, prostate cancer	0–30 μM
↓p-STAT3 and ↓STAT 3 signalling pathway	Breast cancer, hepatocellular carcinoma, pancreatic cancer, prostate cancer	0–50 μM
↓VEGF	Oral squamous cell carcinoma, breast cancer, hepatocellular carcinoma, pancreatic cancer, colon cancer, prostate cancer	0–25 μM
↓IL-6	Hepatocellular carcinoma, pancreatic cancer, prostate cancer	0–25 μM
↑mi RNA	Glioblastoma, breast cancer, lung cancer, pancreatic cancer	0–40 μM
HAT inhibition	Oesophageal cancer, breast cancer	0–50 μM

Bax (Bcl-2 associated apoptosis regulator), Bcl-2 (B-cell lymphoma 2), Bcl-xl (B-cell lymphoma extra-large), NFκB (Nuclear Factor kappa B), MMP2,9 (matrix metalloproteinase 2 and 9), STAT3 (Signal Transducer and Activator of Transcription 3), VEGF (Vascular Endothelial Growth Factor), IL-6 (interleukin 6), miRNA (microRNA).

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
