# Peer review of "Garcinol—A Natural Histone Acetyltransferase Inhibitor and New Anti-Cancer Epigenetic Drug"

_ijms, 2021, doi:10.3390/ijms22062828_

Round 1

Reviewer 1 Report

The paper deals with the garcinol as a natural histone acetylase inhibitor and the new anticancer epigenetic drug. In my opinion, the paper should be rewritten before accepted for publication. Based on the abstract, the reader should expect to find more data about garcinol anticancer properties, the data concerning the mechanism of its anticancer activity based on inhibition of HATs. On contrary, more than the half of the paper is devoted to histone acetylation, HAT enzymes etc. without description or explanation of any connection with garcinol (eg. mechanism of interaction, description of its physicochemical properties…).  Garcinol is mentioned first time in Table 2 as one of HAT inhibitors and then in  Section 3.1. My suggestion is to rewrite the paper and at the beginning give some physicochemical details concerning garcinol, schematic presentations of its mechanism of reaction with crucial enzymes etc.

Reviewer 2 Report

I think generally authors should expand the specific sections and reduce generic background. The tables are really not organized well and understandable.

2nd paragraph. I think the authors should streamline and clarify this paragraph. Do they want to focus only on lysine residues? Because arginine is mentioned 1 time without previous context. The title is histone acetylation not lysine acetylation.

2.1. there is a mistake in the title

Line 74. Even though it’s not the main message of this review. It might be easier for readers to use fewer examples and use full protein names

Reviewer 3 Report

In general, I found the topic of this manuscript interesting although the authors did not add significant additions to the previous body of evidence. Indeed, it seems a summary of previous but recent reviews on the same/similar topic. For this reason, the author should give more information, especially from novel study, to improve the novelty and interest of their work

Round 2

Reviewer 1 Report

The improved version may be interesting for readers of IJMS, and I recommend the publication of the paper in present form.

Reviewer 2 Report

The authors did changes in the manuscript to address the comments and improved the manuscript.

Reviewer 3 Report

I have no additional comments except for those that I have already raised in the previous round of revision.